# Growth Trajectories in Genetic Subtypes of Prader–Willi Syndrome

**DOI:** 10.3390/genes11070736

**Published:** 2020-07-02

**Authors:** Daisy A. Shepherd, Niels Vos, Susan M. Reid, David E. Godler, Angela Guzys, Margarita Moreno-Betancur, David J. Amor

**Affiliations:** 1Murdoch Children’s Research Institute, Royal Children’s Hospital, Parkville 3052, Australia; daisy.shepherd@mcri.edu.au (D.A.S.); niels_vos132@msn.com (N.V.); sue.reid@mcri.edu.au (S.M.R.); david.godler@mcri.edu.au (D.E.G.); angela.guzys@mcri.edu.au (A.G.); margarita.moreno@mcri.edu.au (M.M.-B.); 2Department of Paediatrics, University of Melbourne, Parkville 3052, Australia; 3Department of Clinical Genetics, Amsterdam UMC, University of Amsterdam, Amsterdam Reproduction & Development Research Institute, 1105 AZ Amsterdam, The Netherlands

**Keywords:** Prader–Willi syndrome, weight, obesity, BMI, pediatric, linear mixed models

## Abstract

Prader–Willi syndrome (PWS) is a rare disorder caused by the loss of expression of genes on the paternal copy of chromosome 15q11-13. The main molecular subtypes of PWS are the deletion of 15q11-13 and non-deletion, and differences in neurobehavioral phenotype are recognized between the subtypes. This study aimed to investigate growth trajectories in PWS and associations between PWS subtype (deletion vs. non-deletion) and height, weight and body mass index (BMI). Growth data were available for 125 individuals with PWS (63 males, 62 females), of which 72 (57.6%) had the deletion subtype. There was a median of 28 observations per individual (range 2–85), producing 3565 data points distributed from birth to 18 years of age. Linear mixed models with cubic splines, subject-specific random effects and an autoregressive correlation structure were used to model the longitudinal growth data whilst accounting for the nature of repeated measures. Height was similar for males in both PWS subtypes, with non-deletion females being shorter than deletion females for older ages. Weight and BMI were estimated to be higher in the deletion subtype compared to the non-deletion subtype, with the size of difference increasing with advancing age for weight. These results suggest that individuals with deletion PWS are more prone to obesity.

## 1. Introduction

Prader–Willi syndrome (PWS) is a rare multisystem disorder caused by the loss of transcription of several genes and RNA transcripts on the paternally inherited copy of chromosome 15 [1]. Key early clinical features are infantile hypotonia and poor sucking with failure to thrive, followed in childhood by food seeking and hyperphagia that may lead to morbid obesity [2,3] and early mortality [4] if not externally controlled. Other features are hypogonadism, short stature and small hands and feet. The neurodevelopmental phenotype includes intellectual disability that is usually mild and behavior problems including compulsions, tantrums and skin picking [5].

There are two main molecular classes of PWS. In approximately 65% of cases, PWS is caused by a deletion which removes either 5.0 Mb (class I deletion) or 5.9 Mb (class II deletion) of the paternal chromosome within 15q11-13 [5,6]. In the remaining cases, there is no alteration to the copy number at 15q11-13, and instead the loss of expression from a paternal allele is due either to maternal uniparental disomy of chromosome 15 (matUPD15) or, less commonly, an imprinting center defect (ICD).

Although there is considerable overlap in the clinical phenotypes between deleted and non-deleted subtypes of PWS, several genotype–phenotype correlations have emerged. Individuals with the typical 15q11-q13 deletions have been reported to have lower cognitive ability, including IQ scores, than those with matUPD [7,8,9,10,11,12], whereas the risk for autistic-like behaviors [8,13,14] and the development of mental health issues (e.g., psychosis) [12,15] is elevated in those with the matUPD subtype. These differences all relate to central nervous system function and suggest that there are differences in brain gene expression between the subtypes [10]. The most likely biological explanation for such differences is that the 15q11-13 region contains both imprinted and non-imprinted genes that contribute to the PWS phenotype, with the contribution of non-imprinted genes being limited to deletion cases [16]. For example, the characteristic fair skin in deletion PWS is attributed to haploinsufficiency for the non-imprinted gene *OCA2* at 15q12-13 [17]. Another potential source of phenotypic heterogeneity between deletion and matUPD subtypes is that matUPD15 typically arises from a trisomy 15 conceptus with subsequent trisomy rescue; therefore, the phenotype in some cases of matUPD15 may be influenced by the presence of residual trisomy cells in the placenta and/or the individuals, or by the unmasking of an autosomal recessive gene located on chromosome 15 [18].

Comparisons between the deletion and non-deletion subtypes of PWS provide a means of separating the effects of maternally imprinted genes (which should apply to both subtypes) from those resulting from a single copy loss of non-imprinted genes (which should only apply to deletion cases) [16]. Although growth parameters have been studied extensively in PWS, including the publication of standardized growth charts for children with [19] and without [20] growth hormone treatment, there is limited evidence linking growth to the genetic subtype of PWS. This study aimed to investigate the associations between a PWS individual’s genetic subtype (deletion vs. non-deletion) and their anthropometric measure (height, weight, BMI).

## 2. Materials and Methods

### 2.1. Victorian Prader–Willi Syndrome Register 

This study draws data from the Victorian Prader–Willi Syndrome Register (VPWSR). The VPWSR collects and stores information about individuals with PWS who were born, living and/or receiving services in Victoria, Australia [6]. Ethics approval (RCH HREC ID number 20851) covers the collection of demographic and diagnostic information from medical records for the public good and fulfils state and federal privacy legislation requirements. Families are also provided with the opportunity to consent to receiving questionnaires every 3 years until the age of 18. These questionnaires ask about health, medication use, hyperphagia and other behaviors relevant to PWS. Additional longitudinal data on height and weight are collected for patients of the PWS multidisciplinary clinic of the Royal Children’s Hospital, Melbourne. At the time of this study, the register contained information for 212 individuals with PWS (born in years 1971–2018), with 82 individuals consenting to receive the additional three-yearly questionnaires.

### 2.2. Study Cohort

From the VPWSR, we identified a retrospective cohort of individuals with a known molecular mechanism and at least one recorded height and weight measurement. For each growth measurement, the individual’s age (recorded in years as an unrounded decimal) was calculated based on the patient’s date of birth and the date of measurement as recorded in the register. The study cohort focused on measurements taken from 0 to 18 years of age only, due to the majority of data points falling in this interval. The body mass index (BMI) was calculated from the recorded height and weight measurements (weight (kg) / height (m^2^)). The final data set of interest contained information about: (i) each individual’s genetic subtype (categorized as deletion or non-deletion), (ii) measurements of height, weight and BMI, (iii) each individual’s date of birth, (iv) gender and (v) age at the time of each measurement.

### 2.3. Statistical Analysis

The primary aim of this analysis was to investigate the associations between genetic subtype and anthropometric measures in our cohort of PWS individuals. During the analysis, two different modeling approaches were considered for addressing different questions, as described below. In addition, growth is known to follow different trajectories in males versus females with PWS [20], therefore, analyses were performed for each of the genders separately.

#### 2.3.1. Modeling Longitudinal Growth of Anthropometric Measures

Longitudinal growth models were fitted to estimate the mean growth over time in the observed data and to provide comparison to the normal growth trajectories. An exploratory analysis highlighted a number of characteristics in the data which needed to be addressed when modeling the growth of each anthropometric measure: the non-linear growth over age, the heterogeneity among individuals and the correlation of repeated measures on the same individual. To model the non-linear relationship between age and each outcome, natural cubic B-splines were fitted separately for each genetic subgroup and gender [21]. The spline approach essentially fits separate models to each specified age interval, with the age range divided into intervals through the placement of knots, and provides flexibility in modeling non-linear growth over time. Due to the limited number of observations at older ages, natural cubic B-splines were chosen over unmodified cubic B-splines to improve the stability of the results, particularly beyond the boundary knots [22]. Using natural cubic B-splines allowed for potential non-linear growth over age intervals and generated smooth growth curves for each anthropometric measure by genetic subtype. Age was included as a predictor in the fitted models, with the number and placement of knots (in relation to age) considered during model selection. A combination of the Akaike information criterion (AIC), the Bayesian information criterion (BIC) and visualizations of the raw data and residuals were used to determine the preferred knot placement in the final growth models. In addition, visualizations of the estimated growth curves were used to assess and avoid potential overfitting by ensuring the curves were smooth whilst explaining the signal in the data in comparison to the raw data (i.e., not driven by noise).

To account for the heterogeneity in growth between PWS individuals, linear mixed models (LMMs) with natural cubic B-splines were fitted to include individual-specific random effects in the model. LMMs are a popular choice when modeling repeated measures, due to their ability to account for the imbalance in the distribution of repeated measures for different individuals in the sample [21]. When building the growth models, the inclusion of random intercepts and random slopes were both considered. Random intercepts were fitted to account for the variation in intercept (i.e., height/weight/BMI at birth) for each individual, with random slopes fitted to account for the individual variability in growth over age. When fitting the growth models, we considered both an unstructured correlation structure (assuming each variance and covariance is unique) and a first-order continuous autoregressive correlation structure (assuming measures closer in age are more correlated than measures more distant). The visual inspection of residuals against age alongside variograms were used to determine the most appropriate correlation structure to include in the final growth models.

In addition to point estimates, approximate 95% confidence intervals were estimated to be around the predicted mean growth for each outcome for each subtype, using model-based bootstrap methods for mixed models. Model parameters from the cubic spline models are complex to interpret directly, and therefore these models were used to provide age-specific estimates of anthropometric measures and graphically depict the association between genetic subtype and anthropometric growth in our observed data (without any adjustment), whilst comparing them to the normal growth trajectories. Normal growth information from the Centers for Disease Control and Prevention (CDC) were used in the comparison [23]. These data provided percentiles for the growth of anthropometric measures in children between the ages of 2 and 20 years and were included on the growth charts alongside the growth of the PWS individuals.

For each genetic subgroup (deletion/non-deletion) and gender, the final longitudinal growth models fitted an LMM with individual-specific random effects (random intercept and slope) with natural cubic B-splines (knots at ages 2, 5, 10 and 15 years) and an autoregressive (AR(1)) correlation structure.

#### 2.3.2. Estimating Average Rate of Growth over Age

The previously fitted cubic spline models provided estimates of the differences between mean outcomes for the deletion and non-deletion subgroups across a range of ages in our data (unadjusted). After the visual inspection of the estimated growth curves, we observed the average rate of growth appeared to vary between the genetic subgroups over age. Therefore, we aimed to compare and quantify the average rate of growth over age between the genetic subgroups for each anthropometric measure.

The parameters of the previously fitted spline models do not allow a direct interpretation of the average rate of growth. Therefore, in a second set of analyses, a separate LMM with linear splines was fitted for each outcome. Linear spline models assume a linear relationship between knots, enabling a direct interpretation of fitted model parameters as representing the average growth rate over the interval. The fitted linear spline models included age and genetic subtype as predictors, with an interaction term between age and genetic subtype to allow for different rates of growth between the two subtypes within each age interval. The interpretation of estimated interaction terms provided an estimate of the average difference in the average rate of growth between the genetic subgroups in each age interval.

In addition, each model was explored in two forms: unadjusted and adjusted for the individual’s year of birth (grouped by decade and included as a linear predictor in the model). This adjustment was explored to account for the fact that, compared to the deletion group, data from the non-deletion group were obtained more recently, when different management options were available (e.g., growth hormone availability, the early implementation of food intake restraint). The earliest two decades (1970–1979 and 1980–1989) were grouped together for the adjusted analysis due to no non-deletion individuals being born before 1981 in this cohort. The same model fitting procedure was applied as outlined in Section 2.3.1.

The final models fitted an LMM with individual-specific random effects (random intercept and slope) with linear splines (knots at ages 2, 5, 10 and 15 years) and an autoregressive (AR(1)) correlation structure. Models with fewer knots also provided an adequate fit for the data, with this knot placement was selected to enable the interpretation of the average rate of growth between smaller age intervals and to align them with the previously estimated growth curves.

All of the statistical analyses were performed in R 3.6.1 [24], with the additional packages lme4 [25], nlme [26] and splines [24] used to fit the models, and the ggplot2 package [27] used to produce the growth charts.

## 3. Results

### 3.1. Description of the Study Cohort

The study cohort included 125 people with PWS (63 males, 62 females) with available growth data (Table 1). Of these, 72 (57.6%) had a paternal 15q11.2-q13 deletion, with the remaining 53 (42.4%) categorized as non-deletion (37 had matUPD15; five had an imprinting defect; 11 with other non-deletion). Individuals in the deletion group were born fairly consistently over all decades; however, a large proportion of the non-deletion cohort were born from the year 2000 onwards (83.0%). Of the 125 individuals, 119 (95.2%) had repeated measurements for weight and height over the age range of interest, with an average of 29 observations per individual (range 2–85, median 28). This produced a total of 3565 data points distributed from birth to 18 years of age across the 125 individuals.

### 3.2. Growth in PWS Individuals and Comparison to Expected Growth in the Population

Exploratory data analysis provided insight into the structure and nature of the data and directed the model building process. Spaghetti plots provided visualization of the raw data (Figure 1) with a combination of metrics used to determine the most appropriate growth model. LMMs with the AR(1) correlation structure were the preferred fitted model, with the four-knot model improving the AIC and BIC (see Appendix A for an example of the model fitting metrics). A higher number of knots in the model made a small improvement in the AIC and BIC but appeared to be overfitting the observed data in visualizations of the estimated growth curves, and thus the four-knot model was preferred.

The final cubic spline models were used to generate estimated growth charts, as presented in Figure 2, Figure 3 and Figure 4, with each figure including the comparative quantiles of normal growth range obtained from the CDC. To align with the available growth information from the CDC, the growth charts presented here are restricted to individuals aged two years to eighteen years. Estimates of each outcome (without adjustment) were generated at specific age points (age 2, 5, 10, 15 and 18 years) using the fitted models to provide comparison between the genetic subgroups for each gender (Table 2). It is important to note that there were fewer observations obtained for the oldest age group in both subtypes and genders (18 years of age), and therefore interpretations for the older age group were made considering this.

#### 3.2.1. Height

For females with PWS, individuals with each genetic subtype tended to have a similar growth trajectory until the age of 10 years, after which the rate of growth appeared slower for the non-deletion subtype (Figure 2). At the ages of 10 and 15 years, the difference in height between the genetic subgroups is more pronounced, with the deletion individuals being, on average, 4.6 cm (95% CI: 1.3, 7.9) and 7.9 cm (95% CI: 4.4, 11.3) taller than the non-deletion individuals, respectively. By the age of 18, there appears to be a slight decline in the average height. However, there is high uncertainty here due to measurements obtained from fewer individuals at this time point (three individuals > 17 years of age in the non-deletion group), with two of these specific individuals being the shortest in the cohort and therefore driving the estimations. For males, the growth trajectory was similar between the genetic subgroups, with the estimated difference in mean height being relatively small over different ages (the size of mean difference ranging from 0.4 cm to 2.9 cm; Table 3). Between the ages of 5 and 10 years, there appeared to be a quicker rate of growth for non-deletion individuals. When compared to the normative growth range, the predicted height by age appeared to be lower than the CDC 50th percentile curve for both males and females and both genetic subtypes. For individuals aged 15 years and above, the predicted mean height was even lower than the CDC 10th percentile for both genders and subtypes.

#### 3.2.2. Weight

When looking at weight over age, individuals in the deletion subgroup generally had a higher average weight than those in the non-deletion group for both genders, with the size of the difference varying with age (Figure 2). For females with PWS, the mean weight between subtypes was similar in younger ages (mean difference of 0.1 kg at age two, 95% CI: −4.1, 4.4). However, from ages 7 to 15 years, the difference in mean weight was more substantial, with deletion individuals estimated to be, on average, 10.5 kg heavier than the non-deletion individuals at age 10 (95% CI: 3.7, 17.3) and 8 kg heavier at age 15 (95% CI: −1.7, 17.7). For females with PWS in this cohort, individuals in the non-deletion subgroup had an average predicted weight higher than the CDC 75th percentile, with the predicted weight for individuals in the deletion subgroup even exceeding the CDC 90th percentile for ages of 5 years and above.

The difference in mean weight between the genetic subgroups was less pronounced for males aged 10 years or younger. However, for individuals aged 10 years or above, the mean weight was estimated to be higher for the deletion subgroup. By age 15, the mean weight was estimated to be, on average, 15.7 kg heavier for males with the deletion subtype than the non-deletion subtype (95% CI: −0.8, 32.3). In comparison to the normative growth range, the non-deletion subgroup had an average predicted weight higher than the CDC 75th percentile for males aged 7 years and above, with the average predicted weight in the deletion subgroup being higher than the CDC 90th percentile.

#### 3.2.3. Body Mass Index (BMI)

For BMI, individuals with the deletion mechanism tended to have a higher BMI than those in the non-deletion subgroup, with the size and impact of difference varying with age (Figure 3). For females, the difference was more pronounced in individuals aged 5 to 15 years. At ages 10 and 15 years, females with the deletion mechanism were estimated to have a mean BMI 3.9 units (95% CI: 1.1, 6.6) and 4.8 units (95% CI: 2.0, 7.6) higher than the non-deletion individuals in the cohort, respectively. When comparing to the normative growth range for age four and above, the predicted mean BMI for non-deletion females was higher than the CDC 90th percentile, and even higher than the CDC 97th percentile for deletion females.

This relationship was similar for males with PWS, with the mean predicted BMI being higher in the deletion individuals across all ages. At age five, the mean BMI was estimated to be, on average, 4.9 units higher in the deletion individuals than in the non-deletion individuals (95% CI: 1.6, 8.2). When compared to the normative growth range, the predicted mean BMI for the non-deletion subgroup was higher than the 90th percentile for males aged 7 years and above. For deletion males in the study cohort, the estimated mean BMI in the deletion subgroup exceeded the 97th percentile limit for ages 3 years and above.

Given that the BMI is based on the height and weight of an individual, observations in the growth patterns in BMI can be linked to the other growth charts. For males with PWS, the substantially higher BMI in older ages appears to be driven by a combination of extremes: short stature and high weight in comparison to the normative growth ranges at this age range. For females with PWS, the BMI still exceeds the CDC 90th percentiles but has a less substantial increase in older ages in comparison to the males.

### 3.3. Comparing the Average Rate of Growth between Genetic Subgroups

After exploring the average differences in measures between the genetic subgroups, LMMs with linear splines were used to estimate and compare the average rate of growth between the deletion and non-deletion individuals. Adjustment for year of birth had little to no impact on effect estimates and therefore the unadjusted results are presented here, to allow comparison with the previously estimated growth curves in Section 3.2. Estimates of the unadjusted difference in the average rate of growth are provided in Table 3 (see Appendix A for the adjusted estimates).

#### 3.3.1. Height

The average rate of growth in height was similar for females less than 10 years of age, with the average difference in the rate of growth being relatively small across all ages (Table 3). For males with PWS, the non-deletion subgroup grew at a faster average rate than the deletion subgroup over the ages of 5 to 10 years (unadjusted difference in mean rate of growth = 1.4 cm/year, 95% CI: 0.3, −2.6, *p* = 0.01), aligning with visual observations from the estimated growth curves. However, for males older than 10 years, the deletion subgroup had a slightly quicker average rate of growth (age 10–15: 2.8 cm/year, 95% CI: 1.6, 3.9, *p* < 0.01; and age > 15: 1.8 cm/year, 95% CI: −0.3, 3.9, *p* = 0.09), contributing to similar estimates in mean height between the subgroups.

#### 3.3.2. Weight

For younger females (less than 5 years old), the average rate of growth in weight was similar between the genetic subgroups. During the ages 5 to 10 years, the deletion individuals were growing at a faster rate than the non-deletion individuals, by an additional mean growth of 1.0 kg per year (95% CI: −0.1, 2.2, *p* = 0.08). However, after the age of 10, the non-deletion individuals had a substantially quicker mean rate of weight growth (1.6 kg/year for ages 10–15, and 1.9 kg/year for ages 15 and older). This contributed to the difference in mean weight between the subtypes, reducing in older ages as observed from the estimated growth curve.

For males aged 10 years and younger, the average mean rate of weight growth was similar between the genetic subgroups. However, between the ages of 10 and 15 years, there was a substantially larger rate of growth for the deletion individuals, contributing to the larger mean weight in ages 10 years and above. During this age range, the weight for the deletion individuals was increasing, on average, 3.7 kg per year faster than the non-deletion individuals (95% CI: 2.3, 5.1, *p* < 0.01).

#### 3.3.3. Body Mass Index (BMI)

The average rate of growth in BMI was relatively similar between the genetic subgroups (Table 3). This observation was consistent for both genders. There were some differences in the rate of growth from birth to age two, with the deletion individuals tending to grow at a faster rate than the non-deletion individuals (difference in mean rate of growth in females of 1.4 units/year and 1.1 units/year in males). These estimates were consistent with the growth trajectories observed in Figure 4, with the rate of growth in BMI being fairly consistent, particularly in older individuals.

## 4. Discussion

In this large longitudinal study of 125 individuals with PWS, with analysis using linear mixed models, has shown and described differences in growth patterns between deletion and non-deletion subtypes. The mean weight and BMI were estimated to be higher in the deletion subtype in both genders in this cohort, particularly over the age of 5 years. The mean height was similar in both subtypes for males, with the mean height in deletion females being higher than for non-deletion females over the age of 10 years. For females with PWS, the size of difference in mean weight and mean BMI was more pronounced over the age range of 5 to 15 years, while the difference in mean weight for males was more pronounced over the age of 10 years and fairly consistent over age for BMI. Furthermore, the rate of growth in weight was faster for the deletion subtype in males, particularly in the aforementioned age range (10 years and older). Consistent with published PWS growth charts [19,20], our results show that, compared to the general population, the heights of Australian PWS patients are mostly below the CDC 50th percentile, and weights and BMI are mostly above the CDC 75th percentile.

Our results are comparable to two previous studies of large PWS cohorts aged 16 years and older. Laurier et al. [28] studied 154 PWS adults aged 16–54 years and found a higher mean BMI in the deletion group compared to the non-deletion group (44 vs. 38.9 kg/m^2^). Similarly, Coupaye et al. [29] studied 73 PWS adults aged 16–58 years and found that, compared to UPD patients, deletion patients had a higher mean weight (99.4 vs. 81.0 kg) and BMI (40.9 vs. 34.6 kg/m^2^), but little difference in height. Two earlier studies did not observe a substantial difference in mean BMI between genetic subtypes [30,31]. Studies of neonates [32] and children and adolescents [33] with PWS have also not observed a substantial difference in anthropometric measures between genetic subgroups, suggesting that the differences may emerge with increasing age.

Collectively, these results suggest that deletion PWS patients are more prone to obesity with advancing age, although the underlying mechanisms are not clear. A detailed study of body composition and metabolic profile compared deletion and UPD subgroups but did not observe substantial differences in body fat, adipocyte size, insulin resistance, fasting total ghrelin level or resting energy expenditure, as measured by indirect calorimetry [29]. The examination of endocrine parameters also detected no difference between the two subgroups in GF-1 levels, cortisol levels or the frequency of hypogonadism, but hypothyroidism was more common in the deletion subgroup [29]. Two studies have also examined hyperphagia scores but did not observe a substantial difference in hyperphagia between the two subtypes [29,34]. Yet, it is possible that small differences in hyperphagia or metabolic measures could still lead to a substantial difference in BMI when acting over a timescale of many years. We hypothesize that the observed differences in BMI between the subtypes are most likely central in origin, and may be mediated through differences in gene expression in the brain and associated neurobehavioral phenotypes.

In comparison to the large cross-sectional studies of Laurier et al. [28] and Coupaye et al. [29], a strength of our study was the use of longitudinal growth data obtained from the VPWSR. As a result, a large proportion of individuals in our study cohort (95.2%) had multiple measures for each anthropometric outcome at varying ages, especially up to age 15. The repeated measures enabled us to control for the variability in the outcome measures of the specific individual, and explore the mean growth over a continuous range of ages. As discussed in the methods section, linear mixed models (LMMs) are a popular choice for modeling repeated measures, due to their ability to handle multiple measures on the same individual and robustness to an imbalance in repeated measures for different individuals in the sample. In addition, the use of a first-order continuous autoregressive correlation structure accommodated the within-subject correlations, assuming an individual’s measures at closer ages were more correlated than measures at distant ages. The natural cubic B-spline models provided estimates of growth at varying ages for each genetic subgroup to generate growth curves, with the linear spline models providing estimates of differences in the average rate of growth across subtypes within age intervals. While the assumption of a linear trend under the linear spline model may be slightly simplistic, given that growth is rarely completely linear, the use of such models enabled an efficient estimate of a parameter with a clear and direct interpretation as the average difference between the mean rate of growth across genetic subtypes; an interpretation that is difficult to extract from the cubic B-spline model. Both modeling approaches were useful in addressing different aims, and providing a more detailed insight into the association between genetic subtype and anthropometric measure.

A limitation of this study is that we did not have data about potential confounders or mediators, including potential differences in diet, endocrine factors that are common in PWS [35], physical activity or the use of psychotropic medications. It is also possible that the use of growth hormones may differ between the subtypes, however, it is notable that in Australia, all children with genetically diagnosed PWS receive subsidized growth hormone treatment under the Pharmaceutical Benefit Scheme (PBS) until age 18, unless otherwise contraindicated [36], and without regard to genetic subtype. To account for the varying management of PWS over the years (e.g., growth hormones becoming publicly available in 2011 to those under 18 years of age), an adjustment for an individual’s decade of birth was explored within our linear spline models, although the adjustment had little to no effect on the rate of growth estimates. However, this is essentially a proxy measure using the available data in our cohort and may be limited in completely accounting for the changing landscape of PWS management over the years, which could explain some of the differences between the subtypes observed. In addition, the cubic spline models did not include an adjustment for decade of birth and were used to solely estimate the unadjusted differences observed in our study cohort. Therefore, the estimated difference between subtypes using these models could be due to the deletion cohort being born earlier than the non-deletion cohort. However, based on the minimal effect of adjustment within the linear spline models, we suspect this adjustment would also have a minimal effect in the cubic spline models and the estimates of differences. It should also be noted that there is molecular heterogeneity within each of the subtypes, with the deletion subtype comprising class I and class 2 deletion patients and the non-deletion subtype comprising matUPD and imprinting defect patients, and there may be further differences within these categories that could only be determined by a larger study.

## 5. Conclusions

In summary, in this large sample, we have estimated higher mean weight and BMI in patients with the deletion subtype of PWS compared to patients with the non-deletion subtype. Height was similar for males in both PWS subtypes, with non-deletion females being shorter than deletion females for older ages. Furthermore, the average rate of growth in weight was faster for the deletion subtype in males, particularly in individuals over the age of 10 years. We hypothesize that these differences may stem from known neurobehavioral differences between the subtypes and subtype-specific differences in gene expression.

## Figures and Tables

**Figure 1 genes-11-00736-f001:**
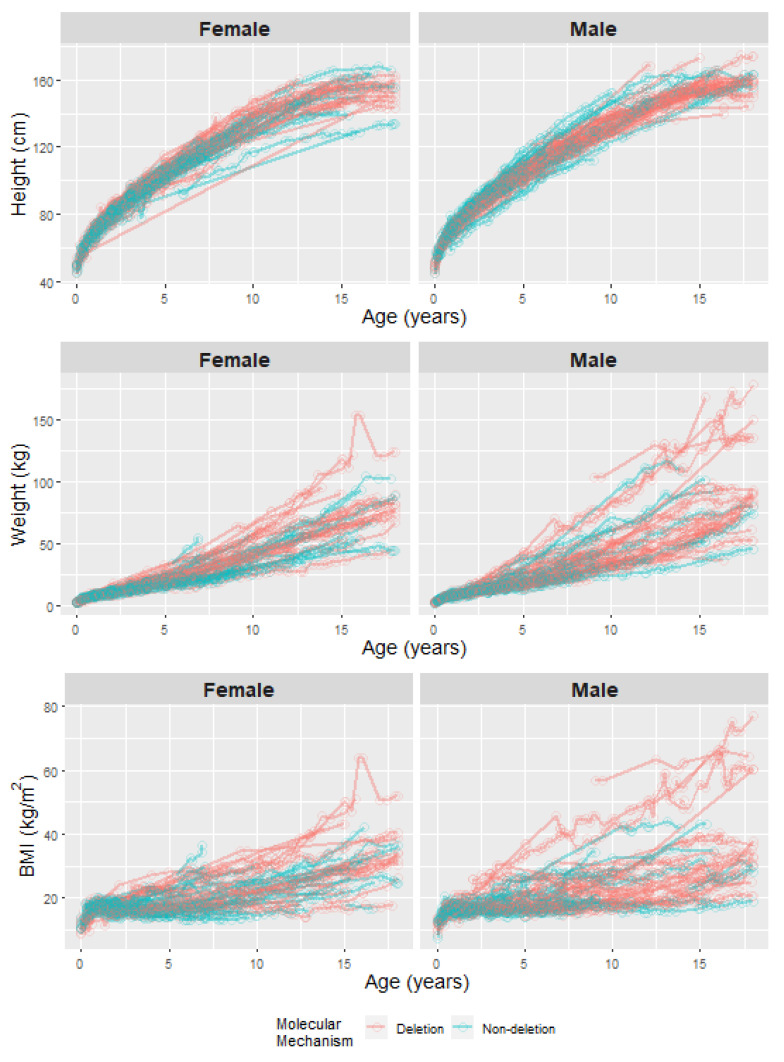
Spaghetti plot of the raw data for each outcome measure from the 125 participants in the study cohort.

**Figure 2 genes-11-00736-f002:**
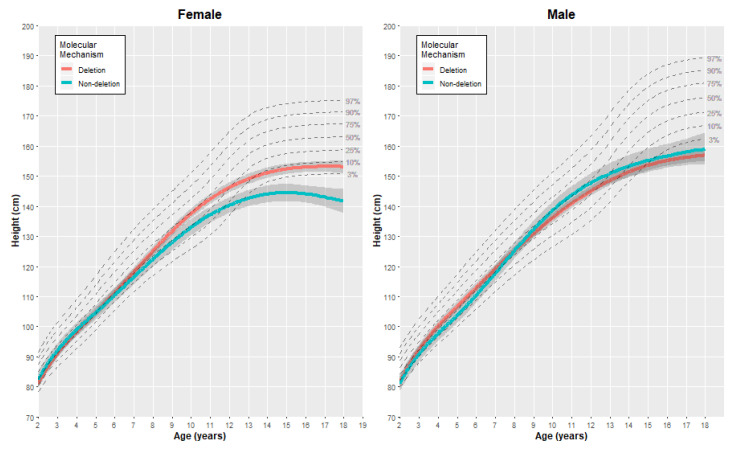
Estimated growth curves for the height of individuals with the deletion and non-deletion mechanism causing PWS in our study cohort (colored lines), alongside the normative percentile ranges of height growth (dashed gray lines).

**Figure 3 genes-11-00736-f003:**
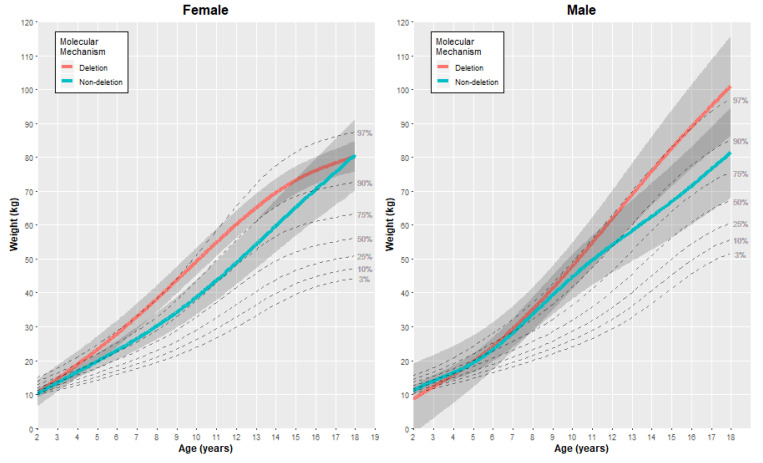
Estimated growth curves for the weight of individuals with the deletion or non-deletion mechanism causing PWS in our study cohort (colored lines), alongside the normative percentile ranges of weight growth (dashed gray lines).

**Figure 4 genes-11-00736-f004:**
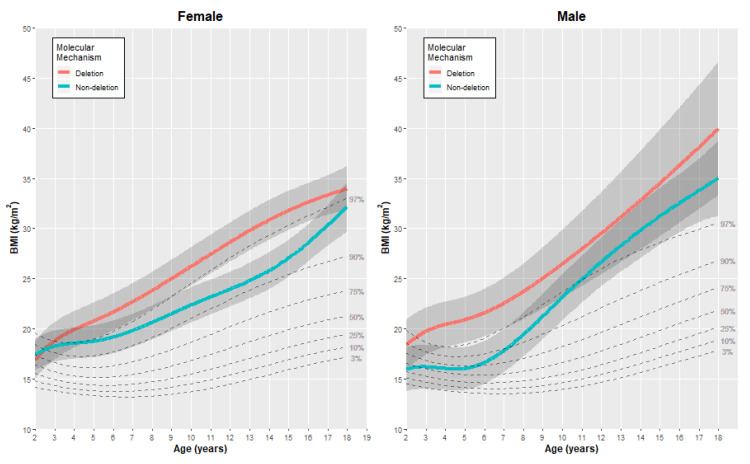
Estimated growth curves for the BMI of individuals with the deletion or non-deletion mechanism causing PWS in our study cohort (colored lines), alongside the normative percentile ranges of BMI growth (dashed gray lines).

**Table 1 genes-11-00736-t001:** Characteristics of the study cohort.

			Genetic Subtype	
			Deletion (*n* = 72)	Non-Deletion (*n* = 53)
Females	*n* (%)		31 (50)	31 (50)
	Number of measurements		915	723
	Age at time of measurement, years–median (IQR)		7.0 (3.2–12.0)	4.7 (2.0–8.9)
	Year of birth–*n* (%)			
		1970–1989	4 (12.9)	1 (3.2)
		1990–1999	13 (41.9)	3 (9.7)
		2000–2009	9 (29.0)	14 (45.2)
		2010–2019	5 (16.1)	13 (41.9)
	Participants with ≥ 1 measurement in each age interval–*n* (%)			
		Birth–2 years	26 (83.9)	26 (83.9)
		> 2–5 years	23 (74.2)	21 (67.7)
		> 5–10 years	24 (77.4)	21 (67.7)
		> 10–15 years	18 (58.1)	10 (32.3)
		> 15 years	15 (48.4)	8 (25.8)
Males	*n* (%)		41 (65.1)	22 (34.9)
	Number of measurements		1225	702
	Age at time of measurement, years–median (IQR)		7.6 (3.7–12.3)	4.8 (2.0–8.6)
	Year of birth–*n* (%)			
		1970–1989	11 (26.8)	2 (9.1)
		1990–1999	12 (29.3)	3 (13.6)
		2000–2009	7 (17.1)	6 (27.3)
		2010–2018	11 (26.8)	11 (50.0)
	Participants with ≥ 1 measurement in each age interval–*n* (%)			
		Birth–2 years	27 (65.9)	20 (90.9)
		> 2–5 years	28 (68.3)	19 (86.4)
		> 5–10 years	32 (78.0)	16 (72.7)
		> 10–15 years	26 (63.4)	9 (40.9)
		> 15 years	21 (51.2)	6 (27.3)

IQR, interquartile range.

**Table 2 genes-11-00736-t002:** Estimates of mean outcome measure by age for individuals with the deletion or non-deletion mechanism of Prader–Willi syndrome (PWS) in the observed data (without adjustment), estimated using linear mixed models with cubic splines.

Gender	Outcome	Age (Years)	Deletion		Non-Deletion		Difference (Deletion – Non-Deletion)	
			Mean	(SE)	Mean	(SE)	Mean	[95% CI]
Female	Height (cm)	2	81.23	(0.81)	82.61	(1.19)	−1.38	[−4.32, 1.56]
		5	104.63	(0.78)	104.62	(1.20)	0.01	[−2.91, 2.93]
		10	137.58	(0.83)	133.00	(1.37)	4.58	[1.31, 7.85]
		15	152.36	(0.87)	144.49	(1.46)	7.87	[4.40, 11.34]
		18	153.11	(1.12)	141.71	(2.04)	11.40	[6.65, 16.15]
Female	Weight (kg)	2	10.46	(1.99)	10.33	(0.63)	0.13	[−4.13, 4.39]
		5	23.25	(1.97)	19.81	(1.15)	3.44	[−1.22, 8.10]
		10	49.26	(2.05)	38.73	(2.62)	10.53	[3.74, 17.32]
		15	73.22	(2.08)	65.21	(4.27)	8.01	[−1.69, 17.71]
		18	80.26	(2.32)	80.73	(5.38)	−0.47	[−12.44, 11.50]
Female	BMI (kg/m^2^)	2	16.88	(0.95)	17.41	(0.80)	−0.53	[−3.07, 2.01]
		5	20.76	(0.94)	18.73	(0.81)	2.03	[−0.50, 4.56]
		10	26.22	(0.98)	22.37	(0.90)	3.85	[1.13, 6.57]
		15	31.80	(1.00)	27.05	(0.94)	4.75	[1.95, 7.55]
		18	34.00	(1.14)	32.18	(1.26)	1.82	[−1.65, 5.29]
Male	Height (cm)	2	81.85	(0.83)	81.41	(1.37)	0.44	[−2.89, 3.77]
		5	106.70	(0.78)	103.78	(1.32)	2.92	[−0.27, 6.11]
		10	136.17	(0.80)	138.58	(1.49)	−2.41	[−5.93, 1.11]
		15	153.60	(0.88)	155.23	(1.96)	−1.63	[−6.10, 2.84]
		18	157.01	(1.05)	159.16	(2.74)	−2.15	[−8.25, 3.95]
Male	Weight (kg)	2	8.75	(5.27)	11.27	(0.66)	−2.52	[−13.57, 8.53]
		5	19.75	(3.89)	19.30	(1.43)	0.45	[−8.17, 9.07]
		10	47.92	(3.51)	44.72	(3.35)	3.20	[−6.89, 13.29]
		15	82.62	(5.73)	66.94	(5.45)	15.68	[−0.77, 32.13]
		18	101.11	(7.54)	81.50	(6.80)	29.61	[−1.51, 40.73]
Male	BMI (kg/m^2^)	2	18.35	(1.28)	15.93	(1.11)	2.42	[−1.10, 5.94]
		5	20.89	(1.16)	16.02	(1.09)	4.87	[1.56, 8.18]
		10	26.49	(1.72)	23.14	(1.17)	3.35	[−0.98, 7.68]
		15	34.53	(2.78)	31.22	(1.45)	3.31	[−3.10, 9.72]
		18	39.95	(3.39)	35.02	(1.90)	4.93	[−3.15, 13.01]

**Table 3 genes-11-00736-t003:** Mean difference in the rate of growth across age intervals, between individuals with the deletion or non-deletion mechanism of PWS, estimated using linear mixed models with linear splines (unadjusted), with an interaction term between age and genetic subtype.

Outcome	Age (Years)	Average Difference in Mean Rate of Growth *(Deletion – Non-Deletion)*					
		Females			Males		
		Estimate	[95% CI]	*p*-Value	Estimate	[95% CI]	*p*-Value
Height (cm/year)	< 2	0.10	[−1.09, 1.29]	0.87	1.35	[−0.20, 2.91]	0.09
	2–5	0.37	[−1.23, 1.98]	0.65	−1.30	[−3.14, 0.5]	0.17
	5–10	0.14	[−0.92, 1.20]	0.80	−1.41	[−2.55, −0.28]	0.01
	10–15	−0.75	[−1.75, 0.25]	0.14	2.76	[1.64, 3.89]	<0.01
	> 15	1.09	[−0.76, 2.93]	0.25	1.82	[−0.29, 3.93]	0.09
Weight (kg/year)	< 2	0.73	[−0.71, 2.16]	0.32	−0.15	[−2.56, 2.26]	0.90
	2–5	−0.36	[−2.14, 1.41]	0.69	0.19	[−2.08, 2.46]	0.87
	5–10	1.03	[−0.14, 2.20]	0.08	−0.23	[−1.67, 1.21]	0.75
	10–15	−1.56	[−2.66, −0.46]	0.01	3.71	[2.30, 5.12]	<0.01
	> 15	−1.87	[−3.90, 0.16]	0.07	1.36	[−1.30, 4.02]	0.32
BMI ((kg/m^2^)/year)	< 2	1.36	[0.58, 2.15]	<0.01	1.07	[−0.11, 2.25]	0.07
	2–5	−1.06	[−2.08, −0.05]	0.04	−0.57	[−1.85, 0.70]	0.38
	5–10	−0.12	[−0.76, 0.54]	0.73	−0.42	[−1.22, 0.38]	0.31
	10–15	−0.38	[−0.99, 0.24]	0.23	0.40	[−0.39, 1.19]	0.32
	> 15	−0.94	[−2.10, 0.22]	0.11	0.20	[−1.27, 1.67]	0.79

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
