# Peer review of "Growth Trajectories in Genetic Subtypes of Prader–Willi Syndrome"

_genes, 2020, doi:10.3390/genes11070736_

Round 1

Reviewer 1 Report

The manuscript presents a longitudinal study of PWS patients (in Australia) who have either a deletion or non-deletion resulting in PWS. They have compared their longitudinal findings  with CDC data for body weight, height and BMI.  Overall the manuscript is well written, but needs a few questions addressed.

  1. The authors mention that there is an imbalance between year of birth deletion versus non-deletion. While one appreciates this information, this implies that it occurred for both males and females. Closer examination of the table 1 data seems to show that the imbalance is only for females.  Would it be possible to comment on this in the response to review, the methods, and perhaps in the discussion? Could this imbalance affect the weight/BMI findings that you had?
  2. Genetically, the CDC is presenting mainly American data, but the PWS patients are of Australian decent. Given that these two countries may represent different ethnicities, especially at the genetic level, the authors might comment on this possibility, or perhaps use Australian control populations (Australian Bureau of Statistics).

Author Response

Comments: Reviewer 1

  1. The authors mention that there is an imbalance between year of birth deletion versus non-deletion. While one appreciates this information, this implies that it occurred for both males and females. Closer examination of table 1 data seems to show that the imbalance is only for females. Would it be possible to comment on this in the response to review, the methods, and perhaps in the discussion? Could this imbalance affects the weight/BMI findings that you had?

Authors’ response: In Table 1 (page 5 line 183) it is seen that there is still an imbalance for males. A small proportion of the male non-deletion subgroup are born prior to 2000 (5 out of 22, approximately 23%), with the majority being born after 2000. Therefore in the male subgroup the non-deletion cohort consists of more individuals born in more recent decades, whereas the deletion subgroup has individuals more evenly spread over the decades. Under the new statistical analysis, estimates of the difference in rate of growth (Table 4; line 289) between the subgroups were adjusted for year of birth, by including this variables a predictor in the mixed models with linear splines. This adjustment was included in these models for all measures in both males and females given there is imbalance in both cases. Given the adjustment, these findings would not be affected by this imbalance, and their agreement with unadjusted analyses with cubic spline models suggest that this factor is having little impact.

  1. Genetically, the CDC is presenting mainly American data, but the PWS patients are of Australian decent. Given that these two countries may represent different ethnicities, especially at the genetic level, the authors might comment on the possibility, or perhaps use Australian control populations (Australian Bureau of Statistics).

Authors’ response: In using CDC growth charts we are conforming to Australian standards for paediatric growth charts (see summary at https://www.rch.org.au/childgrowth/Growth_Charts/). We note that since 2012 the National Health and Medical Research Council of Australia has recommended the use of CDC growth charts in Australian children.

Reviewer 2 Report

This paper presents the analysis of longitudinal growth data collected on individuals with Prader-Willi Syndrome. The availability of such long-term data is a great strength of this paper which can make a meaningful contribution to the PWS literature. However, the statistical analysis needs to be revised in order for the results to be meaningful.

Specific comments

Introduction

  • I am surprised by the statement that half of PWS cases are deletions. Usually the estimate in the literature is higher.

Methods 

  • In mixed models, inclusion of a random intercept and slope accounts for random variation in those parameters between individual subjects. However, to account for correlation between repeated measures on the same subject, the within-subject correlation should be modeled, and the correlation structure used in that modeling should be provided in the methods (e.g., autoregressive structure).
  • I agree with the statement that assuming linear growth between birth and age 18 is an oversimplification. Based on Figures 1-3, linear models would suffer from a systematic lack of fit at certain ages. The authors should provide information about how the fit of the model was assessed and judged to be a good fit to the data. If the model is not a good fit to the data, the results are not meaningful and should not be reported.
  • The linear spline method may provide a better fit to the data, but information should be presented about how the model fit was assessed.
  • A cubic model fitted to each age interval seems prone to overfitting the model to the data. Again, model fit information should be presented. The authors may want to consider fitting one higher order (e.g., cubic) model to the data overall rather than fitting a cubic model in each age interval.
  • Once the authors identify the type of model that best fits the data, the same modeling approach should be used to compare between PWS subtypes and sexes, and to compare the PWS models to published growth curves. The use of different types of models for different purposes is not justified statistically. The authors desire to use simple linear models to more easily compare the groups; however, group-specific estimates and between-group comparisons at specific ages can be obtained from more complex models such as cubic models.
  • Decade of birth would be better to include as a categorical variable (rather than a linear predictor) to account for practice changes over time. While practices have changed over time, the assumption of a linear relationship between decade of birth and growth patterns is probably not valid. Including decade as a categorical variable would allow for more complex associations.

Results

  • Table 1: Please explain what “age” is presented, since each participant is represented in the dataset at multiple ages. It would be more informative to summarize the ages that are covered in the longitudinal dataset.
  • All models should be presented graphically (as in figures 1-3).

Author Response

Comments: Reviewer 2

  1. General comments:

However, the statistical analysis needs to be revised in order for the results to be meaningful.

Authors’ response: We have heavily revised the statistical analysis based on the comments and suggestions of reviewer 2, with Sections 2 and 3 revised accordingly. Highlighted sections in the document indicate where the major revisions have been made.

  1. Introduction:

I am surprised by the statement that half of PWS cases are deletions. Usually the estimate in the literature is much higher.

Authors’ response: In relation to the proportion of PWS cases caused by deletions we have changed this to 65% (line 37) which aligns the references we have used for proportion (Lionti et al. 2015, Butler et al. 2019).

  1. Methods:

(a) In mixed models, inclusion of a random intercept and slope accounts for random variation in those parameters between individual subjects. However, to account for correlation between repeated measures on the same subject, the within-subject correlation structure should be modelled, and the correlation structure used in that modelling should be provided in the methods (e.g., autoregressive structure).

Authors’ response: We have heavily revised the description of the model building process, predominantly in Section 2.3.1. This includes more detail about the random effects fitted in the model, in addition to exploration and determination of an appropriate correlation structure to model the repeated measures (lines 116-129).

(b) I agree with the statement that assuming linear growth between birth and age 18 is an oversimplification.

Authors’ response: We acknowledge that the use of simple linear mixed models is too simplistic for modelling growth curves and is not essential to the discussions within the paper. Therefore we have removed this modelling approach from this revised version.

(c) Based on Figures 1 – 3, linear models would suffer from a systematic lack of fit at certain ages. The authors should provide information about how the fit of the model was assessed and judged to be a good fit to the data. If the model is not a good fit to the data, the results are not meaningful and should not be reported. The linear spline methods may provide a better fit to the data, but information should be presented about how the model fit was assessed.

Authors’ response: Under the revised statistical analysis, we have implemented two modelling approaches – LMMs with cubic splines and LMMs with linear splines (justification expanded on in point 3(e) below). These were built using a thorough model selection process now described in Section 2.3.1 (lines 110-128) for the cubic splines models, and in Section 2.3.2 (lines 164-170).

When interpreting the results in Section 3.2, more information is provided about the model fitting process. Appendix A (line 398) is provided as an example of the model fit metrics when comparing multiple models (random slope only, random slope and intercept, and random effects with an autoregressive correlation structure) with a differing number and placement of knots in the splines model. This demonstrates the AIC, BIC, use of variograms and residual plots when determining the most appropriate model.

(d) A cubic model fitted to each age interval seems prone to overfitting the model to the data. Again, model fit information should be presented. The authors may want to consider fitting one higher order (e.g., cubic) model to the data overall rather than fitting a cubic model in each age interval.

Authors’ response: Whilst fitting the models we were particularly weary of overfitting the data, and therefore considered this during the model selection process. We have updated the manuscript to describe and this process to improve clarity behind the statistical analysis decisions for the reader. Overfitting is discussed in the Methods section (lines 113-116) and the Results section (lines 190-192), with the justification behind using natural cubic B-splines in Section 2.3.1 (lines 101-110).

(e) Once the authors identify the type of model that best fits the data, the same modelling approach should be used to compare between PWS subtypes and sexes, and to compare the PWS models to published growth curves. The use of different types of models for different purposes is not justified statistically. The authors desire to use simple linear models to more easily compare the groups; however, group-specific estimates and between-group comparisons at specific ages can be obtained from more complex models such as cubic models.

Authors’ response: Although the reviewer suggests that different types of models for different purposes is not statistically justified, more recent statistical literature provides a different perspective. For example, discussions in causal inference suggest the use of multiple models tailored to yield different effect estimates (Westreich D, Greenland S. The table 2 fallacy: presenting and interpreting confounder and modifier coefficients. Am J Epidemiol. 2013;177(4):292-298.doi:10.1093/aje/kws412). Indeed, there is a movement away from the naïve view that one can ever construct a single model that reflects the ‘true data generation mechanism’, with instead an emphasis on seeing models as tools to answer specific questions about a system, with different models being suited to different questions. Therefore, under the revised statistical analysis we have included two different modelling approaches. However, we firmly stress that the need for each approach is crucial, and targets different areas when exploring the association between genetic subtype and growth over age.  This separation was a little unclear within the original manuscript, and we hope the new revisions are clearer to the reader.

Firstly, we concerned with modelling the mean of the data in our study for the purpose of describing growth patterns. For this, the longitudinal growth curves for each outcome were modelled using LMMs and cubic splines. Once the most appropriate model was fitted to the data, these were used to (a) generate estimates of mean outcome measure at specified ages by gender and PWS subgroups as well as obtain unadjusted estimates of the mean difference over time, and (b) compare growth in our study cohort to published growth curves.

As evidenced by the estimated growth curves in Figures 2-4, the rate of growth was not constant over time between the genetic subgroups. Therefore in our second question we wanted to investigate this association more closely by obtaining estimates of how this rate of growth differs between the PWS subtypes, while additionally adjusting for year of birth. The previous splines model does not lend itself to such an application, with the model parameters being complex to interpret directly. For this reason, we applied LMMs with linear splines to obtain estimates of the difference in average rate of growth across different age intervals. Naturally we acknowledge that the assumption of linear growth within each interval may be simplistic, but there is a bias-variance trade-off for estimation of the average difference in rate of growth between groups within an interval as specific parameters in these models versus averaging rates within intervals from predictions in a more flexible model. Within the revised paper this is considered in the discussion of our statistical methods.

(f) Decade of birth would be better to include as a categorical variable (rather than a linear predictor) to account for practice changes over time. While practices have changed over time, the assumption of a linear relationship between decade of birth and growth patterns is probably not valid. Including decade as a categorical variable would allow for more complex associations.

Authors’ response: We agree that including year of birth as a categorical variable would be interesting to investigate. However, we have chosen to not include year of birth as a categorical variable for two reasons: (a) issues with model convergence when attempting to include the variable in the model in a more complex manner, and (b) to avoid overfitting the model for this sample size given the current models fitted for analysis. We have rewritten the motivation for each statistical method in the Materials and Methods section (section 2), with description around the adjustment discussed specifically in Section 2.3.2 (lines 147-154).

  1. Results:

(a) Table 1: Please explain what “age” is presented, since each participant is represented in the dataset at multiple ages. It would be more informative to summarize the ages that are covered in the longitudinal dataset.

Authors’ response: The age in Table 1 (line 183) is a summary of the average age at which measurements were taken. We have relabelled this in the table as “age at time of measurement” to improve clarity for the reader. The table alone may not offer a clear idea of the spread of measurements over age.  Therefore, we hope that the inclusion of the raw data plots (Figure 1; line 192) in this revised version will enable the reader to see the distribution of measurements over age for each measure, broken down by gender and genetic subgroup.

 (b) All models should be presented graphically (as in figures 1-3).

Authors’ response: Within the revised version, we have fitted a reduced number of models. The estimated growth charts from the cubic spline models are presented in Figures 2-4. Using the second analysis approach we have fitted linear spline models to obtain estimates for the coefficient of an interaction term only. However, these were used to contribute to the discussion generated from Figures 2-4, with their interpretation referring back to observations in those original three graphs (in section 3.3, lines 273-310). Therefore we do not believe visualizing the linear splines models will substantially contribute to the narrative of the study, and have decided to not present them graphically.

Reviewer 3 Report

The authors described that they investigated the differences in body height, body weight, and BMI between deletion and non-deletion by means of a linear mixed model. Although there was no difference in body height between the deletion and non-deletion, there were significant differences in weight and BMI between them. Although no such differences were found in the reports so far, it is very interesting that the difference was recognized in the studies using LMM.

They need to add their comments against my questions as below.

Why does body height grow linearly after the age of 14 years old in the non-deletion group? Is it the difference between the two groups in the pubertal development? To my knowledge, previous reports have not revealed any differences in it. The authors should add their comments about it.

Why are there differences of body weight and BMI between deletion and non-deletion group, especially in male patients? It may be an interesting idea that the activity due to CNS causes the differences. But is there a difference in the amount of activity that causes the difference in body weight from a young age? Even if there is a difference in an amount of activity between the two groups, is there such a difference in the age when they finally start walking? Has there ever been a report suggesting a difference in the amount of activity? Also, are there genes that account for the difference between the deletion and non-deletion groups? They need to add their comments about that.

Author Response

Comments: Reviewer 3

  1. Why does body height grow linearly after the age of 14 years old in the non-deletion group? Is it the difference between the two groups in the pubertal development? To my knowledge, previous reports have not revealed any differences in it. The authors should add their comments about it.

Authors’ response: (Please note the analysis section shave been heavily revised and therefore response to this comment will reflect the updated Figure 2). In reference to Figure 2, height plateaus in both deletion and non-deletion groups from around age 14 years, and mirrors the plateau in linear growth that is observed in the general population, corresponding to normal menarche. It is presumed that a similar phenomenon is occurring in the PWS population; however, we do not have data on timing of menarche within our population and are not aware of any such data in the literature. We agree that the plateau shown in Figure 2 appears more prominent in the non-deletion group, although this is partly due to the lower height prior to puberty.

  1. Why are there differences of body weight and BMI between deletion and non-deletion group, especially in male patients? It may be an interesting idea that the activity due to CNS causes the differences. But is there a difference in the amount of activity that causes the difference in body weight from a young age? Even if there is a difference in an amount of activity between the two groups, is there such a difference in the age when they finally start walking? Has there ever been a report suggesting a difference in the amount of activity? Also, are there genes that account for the difference between the deletion and non-deletion groups? They need to add their comments about that.

Authors’ response: Other points made by reviewer 3 relate to the mechanisms underlying the differences in weight and BMI between deletion and non-deletion PWS patients. This is indeed a key question arising from our research, and we have highlighted this in the third paragraph of the discussion. Specifically, we have highlighted the results of the two large cross-sectional studies of PWS by Coupaye et al. (2016) and Laurier et al. (2015), which have looked at many variables but found few differences between deletion and non-deletion patients.  We have also added to the discussion further information comparing endocrine findings between the two subgroups (lines 338-341). 

Round 2

Reviewer 1 Report

The authors have responded to the previous reviews with extensive statistical analysis and thorough responses to other questions and concerns. The data on this Australian population will provide additional information on the PWS condition and life trajatory for scientists, clinicians and caregivers.

Author Response

Thank you for this review. No response is required.

Reviewer 2 Report

  1. The revisions to the statistical methods make this a much stronger paper. The authors justify the two sets of statistical models; however, the adjustment for decade of birth in the second set of models but not the first make the two models less directly comparable than is implied in the results section (in comments relating the rates of growth estimated from the second models to the figures generated from the first set). Also see comments 8 and 9. The implications of including decade of birth in only one set of models needs to be addressed.
  2. Methods should specify that decade of birth was included in the models as a linear effect (lines 162-163). This version of the paper does not specify that it was linear.
  3. In Table 1, since the age at all measurements is being summarized, presenting the median age with the interquartile range would be more informative than presenting the mean and SD. Please also report the number of measures being summarized since it will be greater than the number of subjects.
  4. The spaghetti plots are very useful for seeing the distribution of the actual data; thank you for including them. In Table 1 or elsewhere, please report the number of participants, by sex and PWS subtype, with measurements included in each segment of the model as defined by the knots (i.e., how many subjects had data at 0-2 years, >2 to 5 years, >5 to 10 years, etc). That will give the reader a better idea of how sparse the data are for some age ranges, particularly the older ones (i.e., how many subjects had data for ages >15 years).
  5. Results lines 214-217: Please clarify whether there were individuals in the dataset with declining heights, or if the declining height estimated by the model was solely due to there being fewer individuals with measurements at older ages.
  6. Results lines 229-231 and lines 251-252: Statements are made that the deletion subtype tends to have higher weight and BMI, but Table 3 shows that only some of the 95% CIs for those differences exclude 0 (or come close). These statements should be tempered to acknowledge the extent to which the differences may or may not be statistically meaningful.
  7. Throughout the results, please specify that CIs are 95% CIs.
  8. Discussion lines 375-376 says that “an adjustment for an individual’s decade of birth was included within our models” but this is only true for the second set of models comparing growth rates. The Discussion section should include more of a discussion about whether the differences between the deletion and non-deletion subtypes in the cubic spline models could be due to the deletions being born earlier.
  9. The Results and/or Discussion section should also address the implications for interpreting the model results of adjusting for decade of birth in the second set of models but not the first. That would help the reader understand how the two sets of results relate to one another (or do not relate to one another).

Author Response

Comments: Reviewer 2

  1. The revisions to the statistical methods make this a much stronger paper. The authors justify the two sets of statistical models; however, the adjustment for decade of birth in the second set of models but not the first make the two models less directly comparable than is implied in the results section (in comments relating the rates of growth estimated from the second models to the figures generated from the first set). Also see comments 8 and 9. The implications of including decade of birth in only one set of models needs to be addressed.

Authors’ response: We agree that only adjusting one set of models does not make the results as comparable as we initially interpreted them. Therefore, we made the decision to present the unadjusted linear splines output in the results section, with the adjusted table presented as supplementary material. We made this decision based on a number of reasons. Firstly, the adjustment for year of birth provided little to very minimal change on the effect estimates (varying less than 0.5 standard error). Therefore, both the adjusted and unadjusted results were presenting the same conclusions and interpretations stemming from the study. Secondly, by presenting the adjusted estimates it allows for the direct comparison with the estimated growth curves, improving the fluidity of the paper and conclusions from our study. Lines 156-167 have been updated to describe this process and motivation in the methods section, with lines 281-284 justifying this in the results section.

As an alternative approach, we also considered adjusting the cubic splines models. However, presenting age-point estimates using the adjusted model would involve choosing a population to average to. However, this adjustment would mean the growth curves are still not be directly comparable to the linear splines models (due to the selection of a population to average to, which does not align completely with the linear splines parameter estimates). Therefore, we believe the first approach was stronger for allowing the two modelling approaches to be discussed in unison and better suited the discussions within the paper.

  1. Methods should specify that decade of birth was included in the models as a linear effect (lines 162-163). This version of the paper does not specify that it was linear.

Authors’ response: We have updated our description in the methods section to include this specification (line 160).

  1. In Table 1, since the age at all measurements is being summarized, presenting the median age with the interquartile range would be more informative than presenting the mean and SD. Please also report the number of measures being summarized since it will be greater than the number of subjects.

Authors’ response: We have updated Table 1 to (a ) report the number of measures being summarised broken down by gender and genetic subtype, and (b) presented age by median and IQR.

  1. The spaghetti plots are very useful for seeing the distribution of the actual data; thank you for including them. In Table 1 or elsewhere, please report the number of participants, by sex and PWS subtype, with measurements included in each segment of the model as defined by the knots (i.e., how many subjects had data at 0-2 years, >2 to 5 years, >5 to 10 years, etc). That will give the reader a better idea of how sparse the data are for some age ranges, particularly the older ones (i.e., how many subjects had data for ages >15 years).

Authors’ response: We have updated Table 1 to include the number of participants with at least one measurement within each interval (corresponding to the knot placements), broken down by gender and subtype.

  1. Results lines 214-217: Please clarify whether there were individuals in the dataset with declining heights, or if the declining height estimated by the model was solely due to there being fewer individuals with measurements at older ages.

Authors’ response: Within the manuscript we have updated this explanation (lines 215-219). In the non-deletion subgroup in particular, the model is estimated using information for only three individuals over the age of 17. Of these three, two individuals are the shortest in the study cohort and therefore drive the estimation to decline in older ages (i.e., removed the effect from those non-deletion individuals in younger ages but had no measurements at the oldest age point).

  1. Results lines 229-231 and lines 251-252: Statements are made that the deletion subtype tends to have higher weight and BMI, but Table 3 shows that only some of the 95% CIs for those differences exclude 0 (or come close). These statements should be tempered to acknowledge the extent to which the differences may or may not be statistically meaningful.

Authors’ response: We have rephrased some of these interpretations to better reflect the 95% CIs. For example, on lines 234-235 we have rephrased the difference in weight for females to be similar based on the CI being relatively symmetric around 0.  We have also rephrased the general statements (lines 232-235 and lines 254-255) to highlight that the size and impact of this difference does vary with age. In addition, we hope that providing the CIs alongside statements makes it clearer to the reader about how statistically meaningful this is.    

  1. Throughout the results, please specify that CIs are 95% CIs.

Authors’ response: This has been corrected through the manuscript.  

  1. Discussion lines 375-376 says that “an adjustment for an individual’s decade of birth was included within our models” but this is only true for the second set of models comparing growth rates. The Discussion section should include more of a discussion about whether the differences between the deletion and non-deletion subtypes in the cubic spline models could be due to the deletions being born earlier.

Authors’ response: We have corrected this sentence in our Discussion section to reflect the adjustment was made within the linear splines models only (lines 379-382). We have also added to the Discussion and reflected on whether the differences between the subtypes could be due to the deletion cohort being born in earlier years, acknowledging a limitation of our study and reflecting on the adjusted/unadjusted similarities (lines 385-390).

  1. The Results and/or Discussion section should also address the implications for interpreting the model results of adjusting for decade of birth in the second set of models but not the first. That would help the reader understand how the two sets of results relate to one another (or do not relate to one another).

Authors’ response: Based on our comment to comment (1), this explanation is no longer required. We have presented unadjusted results using the second set of models to allow a stronger comparison between the two modelling approaches.